# Stakeholder analysis with regard to a recent European restriction proposal on microplastics

**Lauge Peter Westergaard Clausen**[1]*, **Oliver Foss Hessner Hansen**[1], **Nikoline Bang Oturai**[2], **Kristian Syberg**[2], **Steffen Foss Hansen**[1]

1 Department of Environmental Engineering, Technical University of Denmark, Kgs Lyngby, Denmark,
2 Department of Science and Environment, Roskilde University, Roskilde, Denmark

* lpwc@env.dtu.dk

**Data Availability Statement:** Data is publicly available online. All data can be retrieved from the homepages of ECHA, Ends Europe, Chemical

## Abstract

Stakeholder involvement is pivotal EU governance. In this paper, we complete a stakeholder analysis of the European Chemicals Agency's recent Annex XV restriction proposal process on intentionally added microplastics. The aim of this study is to map the interests, influence and importance of active stakeholders in order to understand the arguments being put forward by different stakeholders and provide recommendations to policy-makers on how to ensure a balanced consideration of all stakeholder perspectives. Stakeholders were identified through niche media analysis and by scrutinising comments from the public consultation on the restriction proposal. Their importance and influence were mapped using three approaches: "scale from low to high", "psychometric scale" and "qualitative ranking". We identified 205 different stakeholders out of which 77 were industry and trade associations, 25 were large companies and only four were small and medium-sized enterprises. National authorities and researchers did not comment on the restriction proposal, whilst large companies were very active providing comments. Industry trade associations and sports-related non-governmental organizations articulated anxiety about the costs associated with the implementation of the restriction proposal. Among environmental non-governmental organizations, there was consensus that plastics should be handled like other substances under EU's chemical regulation. Primary stakeholders identified exhibited high importance, but varying degrees of influence, while the opposite applied to the major European institutions. Based on our analysis, we recommend that: The European Chemicals Agency implement measures to include "silent" stakeholders and invite guest experts to participate in their committees on Risk Assessment and Socio-Economic Analysis; Researchers should be more active in the public consultation; and that special emphasis should be put on helping small and medium-sized enterprises. With regards to stakeholder consultation, we find that media analysis is a good supplement to stakeholder analysis and that a more objective top-down measure of stakeholder importance and influence is needed.

Watch, EURACTICE and EUobserver or through our supporting information.

**Funding:** We acknowledge MarinePlastic, a Danish Centre for research in marine plastic pollution, sponsored by the Velux Foundation, grant number 25084, for providing the funding for this research.

**Competing interests:** The authors have declared that no competing interests exist

## Introduction

Governments across the world have initiated regulatory action and strategies to combat the growing problem of plastic pollution. In Europe, the European Commission (EC) published "A European Strategy for Plastics in a Circular Economy" in January 2018, calling for the European Parliament and Council to sanction a new circular plastic strategy that would present two overall visions for Europe's future plastic economy. First of all, Europe's plastic industry should produce"smart, innovative and sustainable plastics" [1], and second, European citizens, governments and industry should support the sustainable production and consumption of plastics [1]. In the strategy, the EC stated that the transition to a more circular plastic economy required action from all stakeholders in the plastic value chain–from producers, to consumers and to recyclers [2]. The EC therefore called for"*national and regional authorities, cities, the entire plastics value chain, and all relevant stakeholders*", to take"*unrelenting action*" [1]. The circular economy strategy was followed by the release of "A circular economy for plastics" [3], a report presenting a list of recommendations and actions to be taken, with a strong focus on regulatory incentives. One of the policy recommendations was to educate and support citizens, companies and investors on the transition to a circular economy [3]. Another recommendation in the strategy was to develop open collaboration platforms to inform industry, governments and citizens [3]. Most recently, the EU council adopted a conclusion on the transition, in which it is stated that"*The Council calls for action to promote circularity systemically across the value chain, including from the consumer perspective*" [4]. The recommendations in the strategy, as well as the conclusion from the Council, mentioned above all concerns regarding stakeholder involvement, which was viewed as a key component in the transition towards a more circular plastic economy.

Stakeholder involvement is also one of the cornerstones of the EU's governance-based policy strategy [5, 6] and is meant as a way of increasing the efficiency and transparency of regulation [5]. In order for these goals to be met, it is vital that all relevant stakeholder groups are involved in the governance process. The general belief is that those who might be affected by future regulatory initiatives deserve to be involved and have their opinions heard, and furthermore they possess valuable knowledge and information that is relevant when considering the formulation of regulatory initiatives [7]. The successful involvement of stakeholders can additionally secure not only well-contemplated action and smooth implementation, but also the long-term viability of the implemented regulatory actions. Also, stakeholders in civil-societal positions are often better positioned for the early identification of problematic regulatory issues, as they are more directly influenced [8].

The EU governance system has been criticised for not democratically including EU citizens, resulting in the alienation of EU policy [6]. Initiatives like the "Europe for citizens" project have therefore been initiated, in order to encourage and support participation in EU policy-making [9]. However, stakeholder involvement (including citizens and their NGO representatives) is still largely dominated by institutionalised public consultations [10]. In order to gain stronger influence over policy processes, stakeholders tend to form so-called "advocacy coalitions" with other stakeholders that share their interest in a specific topic [11, 12]. Through such coalitions, the individual stakeholder gains a stronger common mandate for influencing the policy process [13], which has been shown to affect EU policymaking processes [14]. The question is, if this formalised inclusion of stakeholders is sufficient to ensure the inclusion of all stakeholders, as the governance framework intends to facilitate [15, 16]. An analysis of how stakeholders communicate and seek influence in policy processes can provide important information about whether all relevant stakeholders are accounted for in technical hearing processes. One strategy that has been used extensively in a diverse set of areas for this type of

analysis, and which is scientifically recognised, is stakeholder analysis (SA) [17, 18]. SA is a decision-support tool, or conglomeration of tools, used to identify, categorise and manage stakeholders from a corporate management, business or regulatory perspective [19]. Ultimately, SA aims to inform decision-makers on involved stakeholders' interrelations, behaviours, agendas, interests and their potential to intervene in the project at hand [20]. SA is normally said to consist of four overall steps: i) stakeholder identification, ii) mapping of stakeholder interests and motives, iii) mapping of stakeholder importance and influence and iv) stakeholder strategy. SA can be conducted either as a bottom-up approach, where stakeholders are involved through structured or semi-structured interviews and surveys, or as a top-down desktop approach, conducted by a research team [19]. The two approaches have different strengths and weaknesses. Where the top-down approach might reflect the biases of the people conducting the analysis and do not actively include stakeholders, thereby missing local knowledge [19], the bottom-up approach suffers from being resource-demanding and time-consuming [21, 22]. Furthermore, although mapping and assessing stakeholder importance and influence are pivotal to SA, Reed and co-authors noted almost a decade ago, that only limited or vague guidance exists on how to do this correctly [19].

In order to review and investigate the pros and cons of top-down desktop SA approaches in general, and methods for assessing stakeholder importance and influence specifically, we started by looking at stakeholder discussions evolving around the European Chemicals Agency's (ECHA) recent proposal for a restriction dossier on intentionally added microplastics [23]. The proposal is a good case to study, as many different stakeholders have already spoken out for and against the restriction proposal (S1–S20 Tables). For instance, the chemical industry branch organisation in the EU (CEFIC) has criticised the microplastics definition used by the ECHA for being"too broad", leaving"room for interpretation" and making the implementation and enforcement of restrictions"challenging" (S3 Table). In contrast, the European Environment Bureau (EEB), an umbrella organisation for NGOs in EU, has argued that the restriction proposal represents a significant step forward, but it strongly warns that the draft law will only restrict cleansing products made by firms that have already pledged to stop using microplastics. Therefore, the EEB argues that the proposal imposes unnecessary delays for most industrial sectors and excludes some biodegradable polymers [24].

The aim of this study is to conduct a SA of the active stakeholders within the regulatory microplastic debate in order to understand the arguments being put forward by different stakeholders and provide recommendations to policy-makers on how to ensure a balanced consideration of all stakeholder perspectives. This includes: (i) identify stakeholders actively participating in the debate on microplastic regulation, including an evaluation of using niche media analysis and scrutiny of a public consultation to identify stakeholders, (ii) to understand their interests and motives and identify possible advocacy coalitions, (iii) to map their importance and influence, to ensure viable future regulatory decisions, including an assessment and evaluation of three approaches for assessing stakeholder importance and influence, and lastly, (iv) to evaluate the overall applicability of top-down SA with the aim of providing recommendations for the further development of SA methodologies.

## Background to the ECHA's restriction proposal on microplastics

In January 2019, the ECHA submitted what is known as a "registration dossier," to restrict the use of intentionally added microplastic particles to consumer or professional-use products of any kind [23]. This action came after a six-month process that started with the European Commission asking the ECHA to prepare an Annex XV restriction dossier–and after a public call for information and stakeholder consultation [23]. The restriction proposal focuses on

"intentional" uses of microplastics, defined in the restriction dossier as follows:""microplastic" means a material consisting of solid polymer-containing particles, to which additives or other substances may have been added, and where $\geq$ 1% w/w of particles have (i) all dimensions 1nm $\leq$ x $\leq$ 5mm, or (ii), for fibres, a length of 3nm $\leq$ x $\leq$ 15mm and length to diameter ratio of >3" [23].

In the proposal, the ECHA specifically proposes the implementation of three types of measures. First is a restriction on placing microplastics on their own or in mixtures on the market, which applies for uses that inevitably result in environmental releases. This measure must be implemented over a transitional period, to allow sufficient time for stakeholders such as the small and medium-sized enterprises (SMEs) and the industry to comply with the restriction. For instance, detergents that contain polymeric fragrance encapsulation, and"leave-on cosmetic products" (such as moisturiser, make-up, lipstick, hair care and hair-styling products), have a suggested 5- and 6-year transition period, respectively, to allow for reformulation and transition to alternatives.

Second, labelling is required from 2021 for items such as construction products, medical products and devices, food supplements, paints and coatings, printing ink and oil and gas, whereby microplastics are not inevitably released into the environment but where residual releases could occur if they are not used or disposed of appropriately.

Finally, reporting requirements apply to any downstream users of microplastics at industrial sites and any importer or downstream user placing a substance or mixture containing microplastics on the market. Specifically, information on polymer identity, a description of the use of microplastics, the quantity of microplastics and the estimated or measured quantity thereof released into the environment have to be reported to the ECHA for the previous year. This information is then compiled by the ECHA and published annually, in order to monitor the effectiveness of the restriction and indicate to organisation if there is a need for further action related to those uses of microplastics that are currently exempted from the proposed restriction [23].

The ECHA believes that these proposed measures will address mixtures that present a risk to the environment, which is currently not adequately controlled, due to the "extreme"–and arguably permanent–persistence of microplastics in the environment contributing to a progressively increasing environmental stock, and future exposures exceeding safe thresholds. Furthermore, ECHA finds that the relevant risk characterisation could be considered in terms of *when* safe thresholds will be exceeded, rather than if this occurs [25].

It is estimated that the implementation of the proposed measures will result in a cumulative emission reduction of approximately 400,000 tons of microplastics over a 20-year period at a cost of approximately €9.4 billion. Quantification is not possible for all sectors, but the average cost effectiveness of avoided emissions is estimated to be €23/kg per year, ranging from €1/kg to €820/kg per year [26].

It is important to note that the intention of the proposed restriction is not to regulate the use of polymers in general but only when specific conditions are met that identify polymers as being microplastics and when their use results in environmental releases. It follows that microplastics formed in the environment as a result of inappropriate and ineffective disposal are considered beyond the scope of the restriction proposal. In the process leading up to the publication of the restriction dossier, many different stakeholders made contributions to the ECHA's call for evidence and engaged in a stakeholder workshop and bilateral discussions. The call for evidence follows a long tradition in the European Union featuring the involvement of interested parties that might be affected by proposed regulatory actions and initiatives [5].

A consultation process on the restriction proposal was furthermore completed by the ECHA at the end of September 2019, in which it repeatedly welcomed specific stakeholder

input, i.e. test methods and pass/fail criteria in relation to clarity, appropriateness, practicality and predictability for assessing the (bio)degradation of microplastics; information to assess the implications of the restriction on granular infill material used in synthetic turf (i.e. granules produced from end-of-life tyres or other synthetic elastomeric materials); the proposed concentration limit of 0.01% weight by weight (w/w) intended to prevent the intentional use of microplastics; the availability of analytical methods that can be used to detect and quantify microplastics in the products above and the presence and concentration of microplastics in a substance or a mixture classed as an impurity [23, 27].

The ECHA's registration dossier is currently subject to a conformity check by the Risk Assessment Committee (RAC) and the Socio-economic Analysis Committee (SEAC), both of which, every eight and ten months, respectively, have to provide their input to the Commission on the ECHA's recommendation [28]. The RAC and SEAC's final opinions are scheduled to be available by March 2020 [29]. Within three months of receiving the opinions of the two committees, the Commission will evaluate the identified risks, benefits and costs of the proposed restriction. If the Commission deems that the identified risks pose an unacceptable risk that has to be addressed on an EU-wide basis, it will provide a draft amendment to the list of restricted substances under the EU's chemical legislation. This draft will then enter a comitology procedure, with scrutiny involving the EU member states and the European Parliament [30]. According to the ECHA [23], the proposed restriction will enter into force in 2021.

## Methodology

As the bottom-up approaches to SA are time-consuming and resource-demanding, they are not appropriate as initial stakeholder screening methods, but they may be applicable at a later stage in the policy process. However, many top-down SA versions have been proposed since Freeman published a key piece of literature in 1984 [19, 31–33]. In this paper, we apply SA as formulated by Rietbergen-McCracken and Narayan [34] in their work from 1998 prepared for the World Bank, as it has a strong policy focus compared to other versions, which are more corporate business-oriented [31, 35, 36]. The approach is a four-step procedure including: Step 1, Stakeholder identification; Step 2, Assessment of stakeholder interests; Step 3, Assessment of stakeholder importance and influence, and lastly Step 4, Formulation of a stakeholder strategy plan [34]. Step 4 is not included in our analysis, as we aim to provide a set of recommendations rather than a strategy plan.

### Stakeholder definition

What constitutes a "stakeholder" or "stakeholders" is key for SA, as it pre-determines who might and might not be identified as being important enough to include in the analysis. The terms have been widely applied with increasing popularity in various contexts since the 1980s [31, 35, 37]. Likewise, numerous definitions have been published on what can be classed as a stakeholder [31, 38, 39], which is why a clear explanation of the term is pivotal in order to know to whom one is referring [40]. In the context of this paper, we use an adapted version of the stakeholder definition proffered by Engi and Glicken [38], namely"An individual or group influenced by–and/or with an ability to significantly impact (either directly or indirectly)–the European Union (EU) regulation of microplastics and which have actively participated in the regulatory debate".

### Stakeholder identification

A variety of methods for identifying stakeholders are available in the literature. These include, but are not limited to, focus groups, semi-structured interviews, snowball sampling, simple

brainstorm activities and media analysis [19, 34, 41]. In this study, we identified stakeholders within the field of microplastics regulation in Europe by scrutinising articles from Chemical-Watch, Ends, EURACTIVE and EUobserver to identify stakeholders publicly speaking out specifically on the ECHA's restriction proposal, as well as on the regulation of microplastics in general. The first published news article on microplastics in the four online media scrutinised dates back to 2013 and our analysis therefore includes articles published in the period from 2013 to September 2019. Hansen and Baun [42] have previously applied a similar method for the use of nano-silver. ChemicalWatch, Ends, EURACTIVE and EUobserver were chosen because they broadly represent specialised niche media and non-profit, independent newspapers, which cover, among other things, the latest trends and developments in chemical and plastics legislation in Europe [43–46]. All four news media were scrutinised for hits on micro-plastic*. The new ECHA restriction proposal for Registration, Evaluation, Authorisation and Restriction of Chemicals (REACH) Annex XV on microplastics was released late January 2019 for public consultation, and it ended late September of the same year. In addition to those stakeholders identified through the analysis of the niche news media, we meticulously went through the comments provided for the restriction proposal, to identify further stakeholders that may not have spoken on the subject in the public media.

All relevant stakeholders were noted, listed and categorised into eight stakeholder groups: National NGOs, International NGOs, National authorities, International authorities, Large companies, SMEs and Industry and trade associations (ITAs) and, Academia and researchers. Furthermore, the stakeholders were categorised as primary, secondary or tertiary according to Lienert [47]. The author defines primary stakeholders as all direct beneficiaries, primary production (industry) and end-users (consumers), whilst secondary stakeholders support or provide services to the primary stakeholders, examples of which include NGOs, researchers and local authorities. Finally, tertiary stakeholders include all governmental bodies, national authorities and international authorities. For each stakeholder group, important representatives were selected for further analysis. The categorisation of stakeholders serves as a mean to secure that stakeholders on all levels are included in the analysis and provides an overview of the role and function of the stakeholders identified [47].

## Stakeholder interests

Mapping stakeholder interests is essential for SA and can be performed in different ways. Montgomery [48], for instance, published a guidance note on SA that included a checklist for drawing out the interests and hidden agendas. The checklist includes a range of self-reflection questions like "What are the stakeholders expectations to the project?" and "What benefits are there likely to be for the stakeholders?". Stakeholders' interests can also be deduced by bottom-up approaches like structured and semi-structured stakeholder interviews and questionnaires [19]. In this study, we determined stakeholder interests by analysing and synthesising individual comments provided on the REACH Annex XV restriction proposal. Furthermore, the opinions and statements of the stakeholders speaking out in the four niche media formats were analysed and synthesised to derive underlying motives and interests. Also, the checklist by Montgomery [48] were partly applied to secure a critical evaluation. In order to provide an in-depth analysis of the stakeholders' interests, the comments and statements were analysed and sorted into five categories: Criteria, Principles, Scientific argumentation, Research needs and Other. The category *Criteria* includes all statements on what the stakeholders would like to have or, in other words, everything related to the "underlying benchmarks" used by the stakeholders to assess or define whether or not, and under what conditions, the proposed restriction is preferred. Examples of these criteria include administrative costs, health benefits

and environmental protection. *Principles* used by stakeholders refer to absolute value preferences that cannot be compromised, e.g. human rights [49]. In the case of the proposed restriction proposal, *Principles* also refer to statements related, for example, to the precautionary principle or other environmental principles. *Scientific argumentations* are statements in which stakeholders refer to scientific opinions, technical insights and academic reflections when voicing their opinions, including making reference to the scientific literature. *Research needs* encompass statements regarding scientific uncertainty specifically, regulatory uncertainty in general and identified knowledge gaps, and finally, *Other* includes all other stakeholder statements and comments of relevance to this analysis.

## Importance and influence

Assessing, mapping and measuring stakeholders' importance and influence is challenging, and the literature only provides vague guidance on how to achieve this goal [19]. The latter especially is difficult, as there exists no commonly applied yardstick [50, 51]. Approaches to assessing stakeholders' or interest groups' influence include process-tracing, assessing attributed influence and establishing differences between stakeholder preferences and the actual policy outcome [51]. However, these methodologies are either bottom-up approaches or backwards analyses, tracing back the initiatives and activities that led to the final policy outcome. As backwards analyses are not applicable for ongoing, incomplete policy processes, and due to the complexity and resource requirements of bottom-up approaches, they are not considered in this study. However, a commonly applied top-down approach is to evaluate importance and influence qualitatively on a scale from high to low [19, 52, 53]. Also, grading the importance and influence on a Likert scale has been applied in various formats, i.e. according to different scales [33, 54, 55]. Lastly, the literature suggests ranking stakeholders in comparison to each other by placing them on a horizontal axis from high to low [56]. In this study, we applied all three approaches, in order to assess their feasibility.

Once the stakeholders' relative importance and influence has been assessed, they can be mapped in an importance-influence matrix, also called the"interest-power matrix" [19, 57]. The matrix is a tool to prioritise stakeholders [19] but also serves as a mean to identify potential vulnerable stakeholders–e.g. stakeholders with a lot at stake but limited power to influence. In theory, the matrix categorises stakeholders according to how they should be managed from a project owner's perspective. Stakeholders with low importance and low influence can largely be ignored, because they cannot affect the strategy plan and will not be affected by it. Stakeholders with high importance and low influence should be "protected," as they have limited means unless they unite in coalitions, but their needs should be met for the initiative to be successful. Stakeholders with low importance and high influence should be continuously considered, informed and monitored, as they could be potentially disruptive for the forming and implementation of the strategy plan. These stakeholders can be managed with minimum effort as long as they remain positive about the project at hand. Finally, stakeholders with high importance and high influence need to be considered key players in the development of the strategy plan, meaning that they have to be actively included and that their needs have to be addressed [58]. The combinations of high and low importance and influence and stakeholder categories are named differently in the literature according to the scope and the context of each project [58–60].

The commonly applied methods for the top-down assessment of stakeholder importance and influence are all qualitative and somewhat subjective [19, 34, 56]. All methods are based on a synthesis and judgement building upon a thorough description and scrutiny of the individual stakeholders, and proxy measures for importance and influence are used in some cases

as justification for the ranking. This might lead to very different analyses and conclusions, and so for this reason, we applied three different approaches to estimating stakeholder importance and influence, and for assessing their feasibility: (i) "Scale from low to high", (ii) "Psychometric scale" and (iii) "Qualitative ranking". In our analysis, importance refers to stakeholders' level of interest in the microplastics regulation. It can also be regarded as denoting "how much" the stakeholders have "at stake". Influence is the perceived power of the stakeholders to affect regulatory decisions, and in this regard their importance and influence were derived from an analysis of who they are, drawing upon the stakeholder identification step and the analysis of their interests and expressed opinions.

The first method we used was labelled "scale from low to high" and involved qualitatively assessing the importance and influence of all identified stakeholders on a low-to-high scale. Loudon and Rivett [61] and Hsu and Lin [62] used this approach in their SA on success in local governance in South Africa and for assessing the benefits of a national park in Taiwan, respectively. However, for both cases, it is not clear how they actually derived importance and influence. Reed and co-authors [19] previously applied this approach for evaluating stakeholder interest and influence relating to a British land-use restoration programme, with the finer point that power was categorised by expert judgement according to Galbraith [63] into sources and instruments of power. More recently, Eyassu [52] applied the approach in her SA on sustainable forest management in Ethiopia. She assessed importance and influence based on a stakeholder description focusing on the stakeholder's role in the project at hand. Hansen and Baun [42] took the method one step further, also using stakeholders' legal prerogative in the EU, total revenue and number of employee as proxy measures of influence. Using this approach in this paper, ranking the importance and influence on a scale from low to high, we adapt the methodology employed by Hansen and Baun [42].

The second method that we used was named the "Psychometric scale," as the importance and influence of the different stakeholders was to be assessed on a psychometric scale based on a review of the stakeholders. This approach is not as prevalent in the literature as the method involving scaling stakeholders from low to high, but it has previously been adapted for evaluating stakeholders' influence on the implementation of construction projects in Sweden [54]. Olander and Landin [54] used a Likert scale ranging from 0 to 10, with 0 indicating no importance/influence, and 10 indicating high importance/influence. Similarly, for a study of stakeholders' environmental influence in the Spanish hotel industry, Céspedes-Lorente and colleagues [55] assessed stakeholder importance and influence on a Likert scale ranging from 0 to 10, with 0 and 10 representing low and high importance/influence, respectively [55]. In this paper, we assessed the importance and influence on a Likert scale ranging from 1 to 10, with 1 as the least important/influential, and 10 as the highest.

Finally, we used "Qualitative ranking" to assess stakeholder importance and influence by ranking them qualitatively, from lowest to highest, against each other. The stakeholder perceived the least important/influential is assigned a value of 1, the second lowest a value of 2 and so forth. This approach has hardly been used according to the scientific literature. However, Padhyoti and colleagues applied the approach in their analysis of the vegetable value chain in Nepal [64]. They based their important-influence rank distribution on the stakeholders' role in decision-making and their importance in the value chain. In our study, we conduct rank distribution based on our analysis of the different stakeholders and their interests.

A common trait of all three methods for assessing stakeholder importance and influence is that they, in this top-down approach, are relying on expert judgement. Thus, they are subjective approaches and reflect the biases of the investigators. It is important to note, that the "scale from low to high" method is a qualitative assessment, whereas the two other methods are semi-qualitative, assigning a number to each of the stakeholders. Using a psychometric

scale may lead to a somewhat ridged analysis, predefining the number of potential importance-influence combinations [54]. During the qualitative ranking approach, the stakeholders are assigned numbers based on their relative importance and influence. In principle, this entails that no stakeholders are equal in importance and influence and that e.g. the most influential stakeholder is located to the far right of the matrix, even though the stakeholder in reality only holds medium to high influence. These limitations and their influence on our analysis are further addressed in the discussion.

## Results and analysis

### Stakeholder identification and interests

In total, 205 stakeholders were identified and then sorted into the following groups: National NGOs, International NGOs, National authorities, International authorities, Large companies, Small- and medium-sized enterprises, Industry and trade associations, and Academia and researchers, as illustrated in Table 1. The largest stakeholder group identified herein is industry and trade associations with 77 registered active stakeholders. National and international authorities represent 21 and 22 stakeholders, respectively, which is similar to the national and international NGOs, with 28 and 17 recorded stakeholders, respectively. Academia and researchers counts 11 active stakeholders, and large companies 25 stakeholders. The smallest stakeholder group is SMEs, which only counts four identified stakeholders. A full list of all identified stakeholders, as well as their comments on the ECHA Annex XV revision and/or their remarks uttered in the niche media, are provided as supporting information in S1–S20 Tables.

The eight stakeholders groups were divided into 28 representative stakeholders based on their interests, as expressed and mapped in step 2 of the SA. The 28 representative stakeholders, their abbreviations, corresponding stakeholder groups and descriptions thereof are presented in Table 2. Also, the categorisation of the stakeholders into primary, secondary or tertiary stakeholders are presented. In total, we identified only two primary stakeholders, large companies and small and medium-sized enterprises, whereas there are 14 secondary and 13 tertiary stakeholders.

With regard to the distribution of stakeholders speaking out publicly in the niche ChemicalWatch, Ends, EUobserver and EURACTIVE media outlets versus stakeholders commenting on the ECHA Annex XV revisions, it is interesting to note that there is great diversity in the data that we collected, as exemplified in Tables 1 and 2 and S1–S20 Tables.

**Table 1. Distribution of stakeholders speaking out publicly in the niche media of ChemicalWatch, Ends, EUobserver and EURACTIVE, stakeholders commenting on ECHA Annex XV revisions and the total amount of stakeholders active in the regulatory debate on microplastics in the EU.**

| Stakeholder | Niche media | Annex XV | Total |
|---|---|---|---|
| National NGOs | 5 | 26 | 28 |
| International NGOs | 13 | 8 | 17 |
| National Authorities | 17 | 5 | 21 |
| International Authorities | 22 | 0 | 22 |
| Large companies | 1 | 24 | 25 |
| SMEs | 2 | 2 | 4 |
| Industry and trade associations | 10 | 69 | 77 |
| Academia and researchers | 11 | 0 | 11 |
| Total | **81** | **134** | **205** |

**Table 2. Representative stakeholders, their abbreviations, corresponding stakeholder groups and descriptions thereof, categorised according to whether they are primary, secondary or tertiary stakeholders.**

| Type | Stakeholders/stakeholder sub-groups | Abbreviation | Group | Description/Examples |
|---|---|---|---|---|
| Primary | Large Companies | L Companies | Large Companies | Consists of all companies with more than 250 employees. |
| | Small and medium-sized enterprises | SMEs | SMEs | All companies with up to 250 employees. |
| Secondary | European Oilfield Speciality Chemicals Association | EOSCA | ITA | Association that responds to regulatory requirements for approval of offshore chemicals and drilling muds. |
| | European Plastics Converters | EuPC | ITA | The trade association of the European plastics converting industry. |
| | International Association of Oil and Gas Producers | IOGP | ITA | The petroleum industry's global forum. |
| | International consumer NGOs | IC NGOs | International NGOs | E.g. the European Consumer Organisation (BEUC) and the Center for International Environmental Law (CIEL). |
| | International environmental NGOs | IE NGOs | International NGOs | Encompasses a range of NGOS including: Beat the microbeat; ECOS; Earthwatch Europe; ChemSec; ClientEarth; WWF, EEB and Greenpeace. |
| | International Pharmaceutical Excipient Council Europe | IPEC Europe | ITA | Global organisation that represents producers, suppliers and end users of excipients. |
| | National environmental NGOs | NE NGOs | National NGOs | National rooted environmental NGOs such as the Scottish Fidra and the German Nature and Biodiversity Conservation Union. |
| | National sport association NGOs | NSA NGOs | National NGOs | Consists of national and regional football associations e.g. Deutscher Fußball-Bund (DFB) |
| | Other national NGOs | ON NGOs | National NGOs | National NGOs whose focus is not directly related to the environment. The group includes: Ellen McArthur Foundation; Breast Cancer UK. |
| | Regional authorities | RA | National authorities | Encompasses municipalities, regional councils and other local authorities. |
| | Researchers | Researchers | Academia and researchers | Includes scientists, college professors and researchers. |
| | Union of European Football Associations | UEFA | International NGOs | The administrative body for football, futsal and beach soccer. |
| | Arctic Monitoring and Assessment Programme | AMAP | International authorities | International organisation that implements components of the Arctic Environmental Protection Strategy. |
| | European Chemical Industry Council | CEFIC | ITA | Main European trade association for the chemical industry. |
| Tertiary | European Chemicals Agency (Including RAC and SEAC) | ECHA | International authorities | Responsible for managing REACH and includes the Committee for Risk Assessment (RAC) and Committee for Socio-Economic Analysis. |
| | European Environment Agency | EEA | International authorities | The EU agency that provides information on the environment. |
| | European Commission | EC | International authorities | The executive branch of the EU, including SAPEA. |
| | European Council | Ecouncil | International authorities | EU body that defines the political direction and priorities of the EU. |
| | European Food Safety Authority | EFSA | International authorities | EU agency that provides scientific advice and communicates on risks associated with food. |
| | European Parliament | EP | International authorities | The legislative branch of the EU, and the 751 elected members from the 28 member states. |
| | International Union for Conservation of Nature | IUCN | International authorities | International organisation working for use of sustainable nature resources in the field of nature conservation. |
| | National elected politicians | NEP | National authorities | Politicians in EU countries. |
| | National environmental governmental bodies (E.g. DK EPA) | NEGB | National authorities | These include national environmental protection agencies as well as forestry, conservation and other governmental environmentally-related bodies. |
| | National governments | NG | National authorities | Governments in Europe. |
| | Nordic Council of Ministers | NCM | International authorities | Body for intergovernmental cooperation of the Nordic Region. |
| | United Nations | UN | International authorities | Intergovernmental organisation responsible for maintaining peace, international relations and international cooperation. Includes UNEP, UNEA and WHO. |

International and national authorities are very outspoken in the niche media, corresponding to more than 25% and 20% of all the comments and viewpoints expressed, whereas their contribution to commenting on the ECHA Annex XV revisions is non-existent for international authorities and limited for national authorities ($\approx$4%) compared to other stakeholders. International NGOs, international trade associations and academia and researchers have been interviewed to an almost equal extend, ranging between 16% and 12%. Academia and researchers have not contributed at all to commenting on the ECHA Annex IV proposal, and international NGOs have only provided a limited contribution overall (5%). Academia and researchers are used to being contacted and interviewed by journalists, but they do not perceive it as part of their responsibility to provide insights and know-how on policy proposals despite the fact that many of the issues and questions raised by the ECHA in the proposal are of a scientific-technical nature (e.g. methods to determine test methods for assessing (bio)degradation, as well as detect and quantify microplastics in products). In significant contrast, ITAs engaged heavily in commenting on the ECHA Annex IV proposal, thus providing more than 50% of all comments received. It is well-known that public consultations are a resource-intensive activity for stakeholders [7], and the heavy involvement of ITAs in public consultations could be a reflection of the fact that they are some of the most resourceful stakeholders. Furthermore, it might indicate that these stakeholders really have something "at stake," that they are influenced by the restriction proposal and they believe that they have an ability to affect significantly the outcome through their participation in the public consultation. Hence, they are very motivated to engage, are very well organised and are able to orchestrate substantial effort in making their views known [65, 66]. The same can be said with regard to large companies, representing 18% of all comments on the ECHA restriction proposal. Interestingly, national NGOs provided almost 20% of all comments in this regard. Here, it is important to note that many of the national NGOs (79%) that have contributed are sports organisations such as Berliner Fußball Verband, Deutscher Fußballbund e.V. and the Spanish Professional Football League (La Liga), each of which has had to answer to the ECHA's call for specific input related to information, in order to assess the implications of restrictions on granular infill materials used in synthetic turf. Large companies and national NGOs are interestingly the only stakeholder groups that have contributed more to comments on the ECHA proposal than they have spoken out in the media. Representatives from large companies have actually spoken out only once, whereas national NGOs have spoken out five times, corresponding to 6% of the overall media contributions made. Here, it is noteworthy that the national NGOs that have spoken out are the Ellen McArthur Foundation, Fidra (an environmental charity based in Scotland), I GLOBAL 200 –Friends of the Earth Austria, DIN-Normenausschuss Bauwesen (NABau) and Naturschutzbund Deutschland (NABU), and hence not the same national sports organisations that were so active during the commenting period. The limited number of SMEs that do provide comments for the restriction proposal express anxiety about the new requirements that they are expected to adhere to as well as for the new technical and administrative burden laid upon them. The stakeholders comments and general opinions promulgated through the media are presented in S1–S20 Tables.

All of the different stakeholders make criteria arguments, some make scientific arguments and only a few make arguments with reference to principles, as highlighted in S16–S20 Tables.

**The use of criteria.** The various stakeholders make considerations regarding a range of different criteria in their comments on the ECHA's restriction dossiers as well as through public media outlets. Criteria where stakeholders have different perspectives include: the status of microplastic under REACH; phasing out and banning intentionally added microplastic; products that should be derogated from the restriction proposal; the cost of obligatory labelling; the

bureaucratic burden for stakeholders; overall societal benefits of the proposed restriction and the optimal length of the proposed transition periods.

Overall, there is consensus among international NGOs that plastics should be approached with the same precaution as other substances under REACH. For instance, the Plastic Soup Foundation writes in their comments on the restriction proposal that "We don't see any reason why polymers, and in particularly plastics, should be treated different than any other chemical substance" (S5 Table). This argument is based on the notion that plastics are very persistent (P), bioaccumulate (B) in the environment and are related to toxic (T) effects, and that the REACH PBT criteria should be applied as the baseline. In contrast, large companies express general concern in their comments about the terms of the restriction and equating all polymers (with special attention to water-soluble polymers) in congruence with other substances regulated under REACH (S16 Table). Specifically, the large companies argue that plastic waste reduction is prioritised over human health benefits, e.g. with regards to pharmaceuticals and healthcare devices (S16 Table). The ITAs support this point, strongly arguing that medical products should be exempted from the restriction proposal.

The large companies also argue that considerable costs are related to obligatory labelling requirements. Along a similar line, the SMEs tend to argue that proposed labelling and reporting requirements place a pointless "technical and financial burden onto them, as they are not able to collect the information stipulated" (S8 Table). The SMEs also argue that some microplastic ingredients have special functions, and hence it can be more difficult for SMEs to change, even though they are well aware of the environmental impacts of microplastics. In their comments, the ITAs question whether the high bureaucratic burden for stakeholders provides any beneficial impact on society and the environment. For instance, the Global Silicones Council argues that the societal consequences associated with the proposed restriction, including disruption to global supply chains and a reduction in innovation, would be extremely disproportionate relative to the current evidence of harm to the environment provided in the restriction dossier (S17 Table). Similarly, the European Crop Protection Association (ECPA) states that the impacts on crop protection and downstream industries have been severely underestimated (S17 Table).

Furthermore, other ITAs question the aim and target of the proposal restriction, arguing that, for instance, cosmetic products should not be regulated if society wishes to prevent microplastics from polluting the ocean, or that only rinse-off cleansing and exfoliating products have been associated with marine litter and not leave-on cosmetics and personal care products (S17 Table). Several national governments, including the Scandinavian, French and British administrations, emphasise their commitment to phasing out and banning intentionally added microplastic in consumer products, especially in cosmetics. Likewise, Belgium has notified the European Commission of a draft plan to voluntarily phase out microplastics in all consumer products by 2019 (S11 Table).

Somewhat along the same lines, academics and researchers have argued in the public media that a ban on single-use plastics will not make a huge difference (S14 Table). For example, Henning Wilts of the Wuppertal Institute for Climate, Germany, said "If you want to have less plastic waste overall, the ban on straws or single-use tableware will only help a little" (S1 Table). In the media, researchers express concern over a number of knowledge gaps. Dr Philipp Schwabl from the University of Vienna, for instance, states that "Now that we have first evidence for microplastics inside humans, we need further research to understand what this means for human health" (S1 Table). Furthermore, many researchers call for the use of the precautionary principle. Dr Anne Marie Mahon, Galway-Mayo Institute of Technology, states in this regard "We don't know what the [health] impact is, and for that reason we should

follow the precautionary principle and put enough effort into it now, immediately, so we can find out what the real risks are" (S1 Table).

The views expressed by the various national NGOs have to be divided into different subcategories, as their views are very diverse. It is clear that the national sport association NGOs, and especially German sports associations (mainly football associations), advocate for an "appropriate" transitional period of at least six years until placing a complete ban on granular infills for use in new plastic turf systems and the conversion of existing surfaces (S12 and S19 Tables). The argument for the long transition period is that the restriction would mean that sports facility operators would no longer be able to acquire the required infill for regular re-filling and that operators would incur costs for changing pitches for which they had not budgeted. In contrast, international NGOs welcome the restriction dossier and find that the transition periods are too long; some even suggest that the ECHA should propose a complete ban, with no lower accepted concentration limits (S10 and S18 Tables). Similarly, the national environmental and other NGOs, in general, speak about the massive plastic production industry and leakage into ecosystems, as well as the potential harmful effects of animal and human health, not only from the plastic itself, but also from additives. Banning intentionally added microplastic in products is not believed to be the solution to this problem; however, it may be a next step towards reducing plastic pollution.

Just as arguments are repeatedly provided by ITAs for additional exceptions to be made, the definition of microplastics used by the ECHA in the restriction proposal is also being accused of being "too broad" by the large companies and ITAs. For instance, PlasticsEurope argues that "We believe that the scope of the proposed definition of microplastics is ambiguous and much broader than the current common understanding of the term "microplastic"."In practice, applying the proposed definition would mean that almost any particle would be considered a microplastic and subject to either a ban or reporting and labelling" (S3 Table).

**Scientific argumentation.**   The different stakeholders use a range of scientific arguments to support their views. Large companies and ITAs tend to call for additional exceptions to be made to the microplastics definition and the scope of the restriction proposal, and furthermore they focus on underlining that there are no sufficient scientific hazard identifications or risk assessments to justify additional reporting or labelling requirements (S2 and S3 Tables).

Several of the ITA scientific arguments provided in the public consultation focus on an individual industry branch and their products, e.g. animal health products, toners for printers, food supplements, medical devices, construction chemical products, film formers and gels and fertilisers, and that their products are already adequately regulated and their usage of polymers should fall outside the scope of the microplastic definition in the restriction proposal (S3 Table). The arguments provided differ but do include that the physical properties of the polymers are permanently modified during use so that they cease to exist as solid particles, that the use of the polymers in commerce will not result in the release of microplastics into the environment and that water-soluble polymers do not exist as particles in the environment, e.g. surface water. Some of the ITAs additionally highlight that changing the composition and formulation of a product might not be as easy as it might seem in the restriction proposal. For instance, replacing a polymer in an approved drug product is an extremely complicated process, the only technical alternatives with equal properties would be paints based on polymers in solutions of VOCs and toners consist of 100% microplastics and thus there is no alternative, since the toner itself meets the definition of microplastics (S3 Table).

The lower size limit of 1nm and the >1% w/w polymer threshold used to define "polymer-containing particles" are questioned especially, and it is argued that it would not be possible to enforce with currently available technical methodologies (see, for instance, PlasticsEurope, VCI and Verband der deutschen Lack- und Druckfarbenindustrie e.V., S3 Table). Several of

the ITAs state that they are not aware of any analytical methods for detection and quantification in medical devices, paints, drugs, films and gels, fragrances, etc. and that the test methods that do exist for analysing particle size distribution and enclosed polymer content cannot be used for the finished product, i.e. in complex and concentrated mixtures. The European Crop Protection Association (ECPA) suggests that the lower size limit should be 1μm, as polymers <1μm are macromolecules, whereas the Bundesverband Korrosionsschutz e.V. suggests that flock fibres up to 15 mm long should not be declared as microplastic (S3 Table). It is also noted that the lack of measurement methods and a size limit that can be easily, accurately and reproducibly measured makes enforcement compliance nearly impossible, and any measurement made would envitably be highly variable (different results for the same polymer in different products) and contestable. Some national environmental NGOs also make an unambiguous call for a harmonised definition of microplastics and standardised testing methods from a somewhat different perspective, as they see these as essential for moving forward regulations and for a possible restriction to be feasible (S7 Table).

Several ITAs argue that polymers are best regulated as substances following a "risk-based approach" and that a quantitative risk assessment can, and should, be conducted for these materials, which is in contrast to the approach taken by the ECHA in the restriction proposal. For instance, the Global Silicones Council finds that it is critical that the restriction proposal includes only those substances that have been evaluated individually based on their characteristics, uses, potential for environmental release and human health and environmental risks (S17 Table). In contrast, other national NGOs emphasise that bioaccumulation and toxicity in animal studies are strong indicators of the risk of general contamination throughout the food chain and across trophic levels–and eventually for human health. For instance, Breast Cancer UK states "There is a growing concern about the impact of human activities on the whole life chain, and there is a legitimate concern that the smaller plastic fraction, through bioaccumulation and trophic transfer, may ultimately contaminate the human population" (S7 Table).

**Principles and research needs.**   Several of the stakeholders refer to principles in their lines of argument; it is stated that the microplastics restriction proposed by the ECHA is different to other REACH restrictions, as it is based on the precautionary principle, due to uncertainty about the hazardous nature of microplastics. For instance, CEFIC states that the precautionary principles are not accurate enough: "The principles that drive the use of the precautionary principle state that the scientific risk assessment should be based on the best scientific data available. The scientific evidence alleged does not meet this standard of evidence required" (S17 Table). It is subsequently argued that placing excessive weight on the precautionary principle will damage the development of those industries using microplastics.

In contrast, some academics and researchers argue that "we should follow the precautionary principle and put enough effort into it now, immediately, so we can find out what the real risks are" (S1 Table). The academic experts call for us to follow the precautionary principle stemming from the numerous research gaps and needs, which still exist but they also stress that increased attention and resources should be given to the allocation of funding for further research (S1 Table).

The ITAs also point out specific research needs, noting that there is a need for an alternative to microplastics and that methods to measure microplastics released into the environment are lacking; consequently, further studies need to be conducted in the subsequent phases, in order to show the release potential into the environment. Specifically, national governments highlight artificial turf pitches as contributors of microplastics into the environment. The Norwegian environmental protection agency (EPA) has opened a consultation on a proposal to prevent the spread of microplastics from artificial turf pitches that use rubber granules (S11 Table). National sports association NGOs stress that available data are highly insufficient, as

no data, risk analyses or impact assessments are available on the released quantities of infills or the effects of any restriction. The development of a better knowledge base should therefore be a first step, before the ECHA imposes the restriction. Further scientific studies are needed in this complex area, in order to close knowledge gaps and to develop more environmentally-friendly materials for sports field construction and to enable an overall assessment of existing plastic turf systems based on sustainability criteria.

National governments also seek further research into hazardous properties, substitution and the prevention of leaks into the environment of added microplastics. Netherland's National Institute for Public Health and the Environment (RIVM), for instance, states that it is essential to create awareness of the problem among consumers and professionals. It argues that the release of microplastics can be reduced through innovation and by implementing necessary measures (S11 Table). Along these lines, some international NGOs require the establishment of criteria for the (bio)degradation of plastics that reflect the variety of environmental matrices in which plastic particles gather (S5 Table).

## Other arguments

Some stakeholders make use of other arguments that are not related to criteria, science or principles and research needs. These other arguments are more related to values and sentiments or economic and administrative burdens. E.g. several NGOs accuse ECHA for "unduly limiting" the scope of the proposal by taking into account the agendas of the industry before consulting its scientific committees and for being subjective in their work favouring the industry (S5 Table).

The German sports NGO argue that a ban of intentionally added microplastics will have severe economic consequences for their member clubs, leading to a marked loss of welfare (S10 Table). It is also pointed out by the ITA that the vast majority of companies, especially SMEs, do not have the required equipment and personnel to prepare the proposed annual reports outlined in the restriction proposal or to measure according to the proposed definition, and it is not clear that enforcement authorities and official control laboratories also have access to the required equipment (S3 Table).

## Stakeholder importance and influence

Stakeholder importance and influence were mapped according to whether we perceived them as being "high", "medium" or "low," according to our judgement, on a psychometric scale and according to a qualitative internal ranking from high to low, Fig 1. In general, the large European governmental bodies, namely the European Council, Parliament and Commission, are perceived to have a high influence, as they are all positioned in the upper echelons of the European Union. To exemplify this point, the Parliament and the Council are the law-making institutions in the EU, and the Commission is the executive body responsible for proposing new legislation [2], and it was the Commission that originally asked the ECHA to start preparing the dossier [67]. Overall, they are all considered to be of high importance with respect to the restriction proposal. However, at this stage of the proposal process, the Parliament and the Council are deemed to be of medium importance.

We have placed stakeholders such as the ECHA in the upper-right corner of the importance-influence matrix (high importance, high influence), in Fig 1, as we find them to be key players in the restriction process. The ECHA is perceived to be highly important and influential, as it prepared and eventually published the restriction proposal, managed the public consultation and revised the restriction proposal according to its evaluation of the merits of received comments [68].

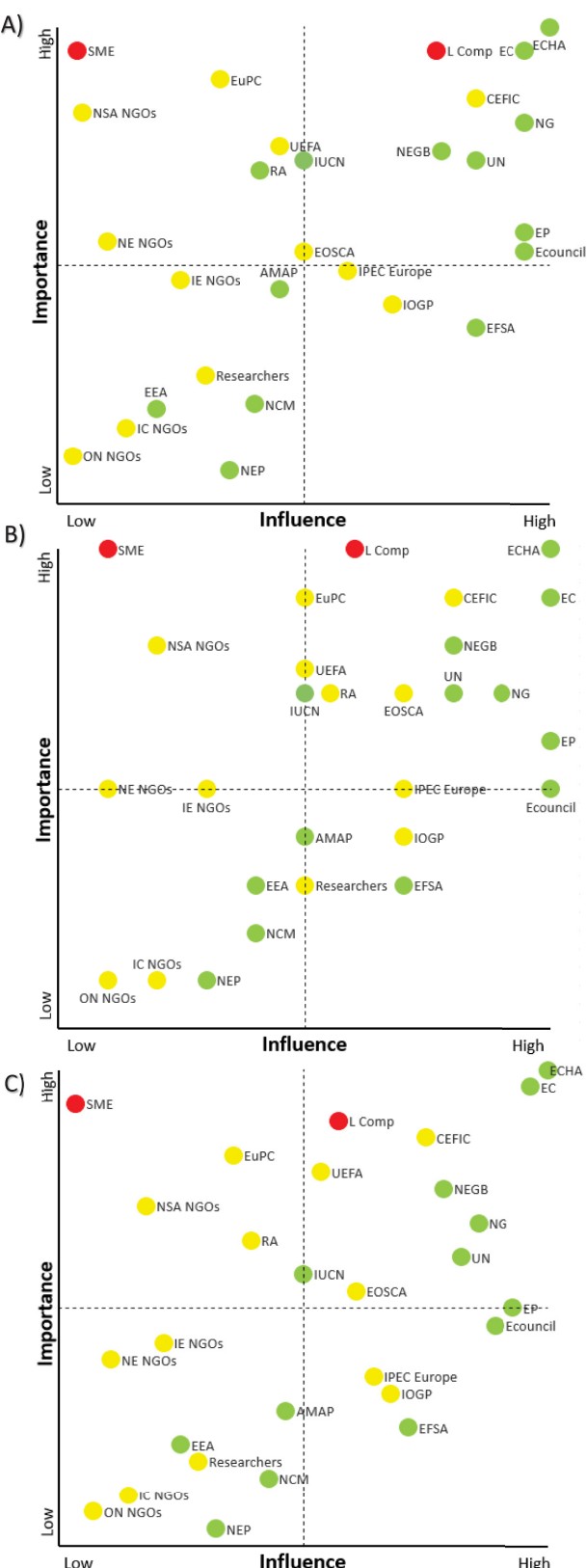

**Fig 1.**  Impotrance-influence matrices constructed by three different qualitative judgement approaches: a) Scale from low to high, b) Psychometric scale and c) Qualitative ranking. Primary, secondary and tertiary stakeholders are marked red, yellow and green, respectively.

Other stakeholders placed in the same corner include the UN, national governments and their environmental bodies. The UN is not directly influenced by microplastics regulation but has nevertheless voiced strong opinions and concerns about their presence in the environment; for example, Jacob Duer, director of the chemicals and health branch of the UN Environment Programme, said that "plastic waste is emerging as one of our greatest environmental challenges" (S4 Table). Due to their leading positions and highly esteemed scientific advice [69], they are perceived as having high influence. National governments play a limited role with respect to restrictions under REACH, although they do have influence on the restriction proposal itself during the formulation process. For instance, during the informal risk management and evaluation (RiME+), meetings were co-organised by the hosting member state competent authority (MSCA) and the ECHA three times a year, in order to facilitate voluntary coordination and non-binding discussions between the individual member states and the ECHA on activities related to the implementation of the integrated regulatory strategy, covering the different REACH/CLP processes. The national governments and their environmental bodies are subject to the new restriction proposal, but the governments are at the same time in a position to challenge the European commandments–albeit at the risk of being taken to court.

Also, in the same corner of the importance-influence matrix we find CEFIC and large companies. The large companies are subject to the new restriction proposal and thus have a lot at stake (high importance). Large companies are considered to have high influence, as they typically have a quite substantial economic basis for lobbyism [70]. For instance, BASF generated in 2018 a total revenue of approximately €63bn and employed more than 122,000 people [71], which means it had a substantial economic surplus and carries considerable societal weight with regard to employment and influencing political decisions–as also reflected in their comments on the ECHA Annex XV restriction proposal. In general, CEFIC represents industry and not only large companies, and so its importance and influence are therefore somewhat related. CEFIC is deemed to have a slightly higher importance, as it also represents the SMEs, which have a lot at stake (very important) but are much more vulnerable. SMEs do not have the same economic power as the larger companies. This is reflected by the fact that only four SMEs are represented among the active stakeholders in the microplastics regulation debate in Table 1, and consequently they are considered to have low influence.

Together with the SMEs in the upper-left corner of the matrix are the national sports NGOs, which are vulnerable to the restriction proposal and have to be "protected" (high importance, low influence). As the restriction proposal includes the use of artificial turf, it will have consequences for the national sports NGOs, and so their importance is judged as being medium to high. Their influence is deemed low, as they have to rely on politicians to pick up their agendas. Dür and De Biévre [72] point out that NGOs, in general, have moved much closer to policymakers, but despite this they have not had the power to influence the policy outcomes. Also, following the same argumentation, the national environmental NGOs are have low influence and the international environmental NGOs low to medium influence. Both of them are considered medium in importance, as it is generally essential to include them in public consultations, in order to foster a more participatory democracy [73] despite the fact that they are not very resourceful when it comes to manpower and funding.

At the upper-centre of the matrix we placed a cluster of stakeholders encompassing the Union of European Football Associations (UEFA), the International Union for Conservation

of Nature (IUCN) and regional authorities. With an estimated revenue for 2019/2020 of more than €3bn [74], UEFA do have some influence, but the money to a certain extent is earmarked for sport and less for lobbyism. With respect to their importance, UEFA is considered medium to high, based on the notion that many of their members responded actively to the ECHA's call for input into the restriction proposal, which again indicates that they are resourceful and well organised. IUCN is a union made up of more than 1,300 civil and government organisations [75], and it is considered medium in influence but medium to high importance, due to its active role in the debate. Regional authorities have little impact directly on the restriction proposal but will be affected by it, as they are in an executive and informative position. Therefore, their importance and influence are deemed medium to high and medium, respectively. Also located centrally in the matrix, but slightly lower in importance than UEFA and IUCN, is the Arctic Monitoring and Assessment Programme (AMAP). The role of AMAP is solely to monitor, document and provide science-based information to decision-makers [76], and their importance and influence is judged as medium.

The oil-related industry associations the European Oilfield Speciality Chemicals Association (EOSCA) and the International Association of Oil and Gas Producers (IOGP) are both considered medium in importance, as they are indirectly affected by the restriction proposal. Furthermore, they are believed to be medium to high in influence, as they represent influential industries with an economic interest. European Plastics Converters (EuPC) is judged to be high in importance, as its members are directly influenced by the restriction proposal. However, its influence is not as profound (medium) as the oil industry, even though it is still significant.

The European Food Safety Authority (EFSA) and the International Pharmaceutical Excipient Council Europe (IPEC Europe) are both considered medium to high in influence: EFSA due to its influential position as one the European Commission's scientific agencies, and IPEC Europe due to the economic power of the pharmaceutical industry. However, their importance is currently considered low to medium, as they are only indirectly involved and affected by the restriction proposal. However, this might change over time if plastics become a food safety issue.

The Nordic Council of Ministers is known to be proactive with respect to environmental issues. On behalf of the ministers, Guðmundur Ingi Guðbrandsson, Minister for the Environment and Natural Resources of Iceland, proclaimed "The Nordic region must be a pioneer in reducing the environmental impact of plastics. . ." (S9 Table). Their importance and influence are perceived low to medium and medium, respectively, and they are thus situated in the lower-central part of the matrix together with national elected politicians, EEA and researchers. In the lower-left corner, stakeholders with relatively low importance and influence are located. These include other national NGOs and international consumer NGOs, as they are not directly involved in or affected by the restriction proposal, and they do not have means to directly influence it. However, the fact that they have been active in the microplastics debate justifies that they are part of the SA. On the foundation of these arguments, and based on the three different assessment approaches, assessments of the stakeholders' importance and influence are presented in Table 3.

Observing the results of the three approaches for mapping stakeholders' importance and influence, as shown in Fig 1, it is evident that all of them yield almost identical results and capture the same trends. It is important to note here that the primary stakeholders are considered high in importance, albeit with varying degrees of influence. Conversely, the major European institutions (half of the tertiary stakeholders) are all considered high in influence but with varying degrees of importance. The secondary and tertiary stakeholders do not follow any specific trends but are characterised by being scattered throughout the plots.

**Table 3.  Assessed stakeholder importance and influence according to the following three applied methodologies: Scale from low to high, psychometric scale and qualitative ranking.**  Values given are importance/influence. Stakeholders are listed in alphabetical order.

| Stakeholder | Scale from low to high | Psychometric scale | Qualitative ranking |
|---|---|---|---|
| AMAP | Medium/ Medium | 4/5 | 8/13 |
| CEFIC | High/High | 9/8 | 24/21 |
| EC | High/High | 9/10 | 27/27 |
| ECHA | High/High | 10/10 | 28/28 |
| Ecouncil | Medium/High | 5/10 | 13/25 |
| EEA | Low to medium/Low to medium | 3/4 | 6/7 |
| EFSA | Low to medium/Medium to high | 3/7 | 7/20 |
| EOSCA | Medium/ Medium to high | 7/7 | 15/17 |
| EP | Medium/High | 3/10 | 14/26 |
| EuPC | High/Medium | 9/5 | 23/10 |
| IC NGOs | Low/Low | 1/2 | 3/4 |
| IE NGOs | Medium/Low to medium | 5/3 | 12/6 |
| IOGP | Medium/ Medium to high | 4/7 | 9/19 |
| IPEC Europe | Medium/Medium to high | 5/7 | 10/18 |
| IUCN | Medium to high/Medium | 7/5 | 16/14 |
| L companies | High/High | 9/8 | 25/16 |
| NCM | Low to medium/Medium | 2/4 | 4/12 |
| NE NGOs | Medium/Low | 5/1 | 11/3 |
| NEGB | Medium to high/Medium to high | 8/8 | 21/22 |
| NEP | Low/Low to medium | 1/5 | 1/9 |
| NG | High/High | 7/9 | 19/24 |
| NSA NGOs | Medium to high/Low | 8/2 | 20/5 |
| ON NGOs | Low/Low | 1/1 | 2/2 |
| RA | Medium to high/ Medium | 7/5.5* | 18/11 |
| Researchers | Low to medium/ Low to medium | 3/5 | 5/8 |
| SME | High/Low | 10/1 | 26/1 |
| UEFA | Medium to high/Medium | 7.5*/5 | 22/15 |
| UN | Medium to high /High | 7/8 | 17/23 |

* Corrected by 0.5 to avoid any overlap in the matrix mapping.

## Discussion

As stakeholder involvement is considered one of the cornerstones of EU governance strategy [5], it is vital that all relevant stakeholder groups are either involved in the governance process or at least have been given a chance to participate. With respect to this, and to the ECHA's Annex XV revisions, our analysis show several interesting aspects including, among others, how important stakeholders are not included in the Annex revision process and how stakeholders form advocacy coalitions.

### Missing out on important stakeholders in public consultations and media

Interestingly, the ECHA Annex XV restriction proposal process for inclusion of stakeholders did not capture all prominent stakeholders. E.g. national authorities and researchers did not comment on the restriction proposal during the public consultation organised by the ECHA, and international authorities provided very few comments. In contrast, they were very outspoken in the media, possibly because international authorities, as defined in this study, include many European institutions that would normally be expected to have provided comments

internally. Conversely, large companies were very active in providing comments on the restriction proposal, but they did not speak out in the media.

Recently, Junk [73] showed that public consultations can provide less resourceful stakeholders, e.g. NGOs, institutional access, thus ensuring the stronger participation of specific local or sectoral interests. A less direct, but still important, mean of obtaining influence can be obtained by speaking out in the media. Thus, a scrutiny of the media is a good addition to the stakeholder analysis step of the SA, which also fits well in the further analysis of stakeholders' interests. More resourceful NGOs are able to influence the media discussion; however, the less prominent ones are not so, and they are only weakly represented in media debate [73]. We do not see a similar trend in our analysis as the most resourceful NGOs (the sports NGOs) did not speak out in the media but where heavily represented in the public consultation.

Surprisingly, SMEs were hardly identified in any of the two analyses. The analytical comparison illustrates that some stakeholders, which are active in the media are not represented in the public hearing. This implies that these stakeholders could be neglected if inclusion of stakeholders in the policy process is solely based on the public hearing.

When conducting a SA, it is important to realise that step 1, i.e. stakeholder identification, is a restrictive part of the analysis, because if the identification process fails to identify specific stakeholders, they are excluded from the rest of the analysis, potentially undermining the bedrock of the SA. An obvious shortcoming of the SA conducted herein–and maybe the ECHA's public consultation process itself–is that it fails to identify the "quiet" or "silent" stakeholders such as the general public, which has neither provided comments on the ECHA's Annex XV restriction proposal for intentionally added microplastics nor spoken out in the niche media surveyed in this study (S1–S20 Tables). If, and when the restriction proposal enters into force [77], the public will be affected, as specific consumer products will cease to be available. The present analyses illustrate that different approaches to identifying stakeholders yield different results, and it might therefore be feasible to use a suit of approaches to identify stakeholders rather than relying on the institutionalized public hearing alone. Proximity and fostering close partnerships on a local scale are crucial for a future "citizen-owned" European Union [6]. However, if the citizens are not heard or included in the process, regardless of whether they want to be included or not, it is challenging to establish such a partnership and uphold trust. In the current situation, the public has to rely on other stakeholders (politicians, national and international authorities, NGOs, companies, etc.) to promote their interests, whatever they may be. This raises the question whether the public consultation is meant for the public or serves instead as a justification for stakeholder involvement. A stakeholder group poorly captured by the public consultation and by the scrutiny of the niche media is SMEs, as they account for just four out of the 205 identified stakeholders. In Europe in 2015, over 99% of non-financial business entities were SMEs, accounting for 23.4 million companies [78]. From this perspective, it is alarming how seemingly underrepresented the SMEs are in the debate, and clearly initiatives should be launched to include them more actively in the future. Another group of stakeholders seemingly only contributing to a limited extent to the public consultation is academics and researchers. Of course, scientific comments and input might be provided from both the ECHA's Committee for Risk Assessment (RAC) and the Committee for Socio-Economic Analysis (SEAC), but the fact that academics and researchers have not commented on the restriction proposal, and were identified solely through the media analysis, is disturbing. Further, looking into the members of RAC and SEAC, RAC have no members, which are experts on plastics or microplastics and SEAC only holds one member with expertise in polymer chemistry and physics [79, 80]. In the case of microplastics, it is palpable that there is a lack of relevant specialised expertise, which might undermine the scientific advices provided to ECHA. We argue that a solution might be to invite 3–5 guest experts with plastic-specific

knowledge to each of the committees. For instance, the Commission's Scientific Committee on Emerging and Newly Identified Health Risks has previously applied this approach [81].

## Stakeholder interests and coalitions

When commenting on the restriction proposal, stakeholders often use arguments based on their own sets of criteria. Scientific arguments are also frequently applied, but only a few stakeholders make arguments relating to principles. Looking in more detail at the criteria referred to in the stakeholders' expressed opinions, industry and their associated trade associations often articulate anxiety, due, for instance, to the cost and the bureaucratic burden of mandatory labelling and the overall societal benefits of the restriction proposal. These concerns are supported by sports-related NGOs, which advocate for a long transitional period of phasing out artificial turf. With respect to artificial turf, national governments are strongly in favour of a ban, expressing concern over the possible environmental release of microplastics. Among the international NGOs and environmental NGOs, there is consensus that plastics should be handled like any other substances under REACH, and they welcome a ban of intentionally added microplastics. Additionally, environmental NGOs call for attention to and more research on the (bio)degradation of plastics (S1–S20 Tables).

Grounded in the analysis of what has been said in the niche media and in the comments on the restriction proposal, stakeholders can be separated into those who are in favour of the restriction proposal, those who are against it and those who have a neutral or objective attitude towards it. From an overall perspective, the national and international authorities are predominantly in favour of the restriction proposal. An exception is EFSA, which is more reluctant, stating "Currently, there is no evidence to suggest that there is a food safety risk" (S4 Table). Contrarily, industrial-related stakeholders are against the restriction proposal, whilst the perceptions of national and international NGOs are more diverse. Here, a pronounced line is drawn between environmental and consumer NGOs and sports NGOs, with the latter not in favour of the restriction proposal.

It is well-established that stakeholders form coalitions in order to support or oppose specific policy options [82]. Research completed by Klüver [65] indicates that large coalitions are more influential than small, separate groups. Based on a review of characteristics on successful lobbyism across 56 policy issues, including 2,696 interest groups, Klüver noted that stakeholders that are part of a large coalition have more salience and a greater impact. Therefore, there is a significant advantage for stakeholders in forming alliances dependent on shared interests. With respect to this point, an interesting observation is that the arguments brought in play by the sports NGOs, especially the German football associations, seem highly coordinated, as they provide identical arguments and support the claims by industrial-related stakeholders that the restriction proposal will have severe negative social and economic effects (S1–S20 Tables). We believe that this indicates that there is an "advocacy coalition" between industry-related stakeholders and the national and international sports NGOs, building upon a shared interest in opposing the restriction proposal. This type of coalition has previously been observed in the context of European chemicals regulations, when animal welfare organisations entered into an advocacy coalition with industry to stop animal testing [83]. The immense amount of identical comments made by German sports NGOs (S19 Table), and the overwhelming interest from industry-related stakeholders [84], might indicate a disruptive attempt to overburden the consultation process, in order to slow down the restriction process.

## Methods for evaluating stakeholder importance and influence

Evaluating stakeholder importance and influence is a key aspect of any SA [34]. Methods for evaluating stakeholders are available in the literature, and we used three of these in this study

to test their overall applicability, namely scaling from low to high, psychometric scale and qualitative ranking. The three approaches applied for mapping the stakeholders' importance and influence in matrices captured the same overall trends. The primary stakeholders have high importance but varying degrees of influence, whilst the opposite applies to major European institutions, which are high in influence with varying degrees of importance.

One of the major differences between scaling from low to high, qualitative ranking and using a psychometric scale is that using a psychometric scale from 1 to 10 gives a somewhat rigid analysis with limited flexibility to reflect differences in stakeholder importance and influence. The psychometric scale provides only 100 different combinations of importance and influence, meaning that with a high number of stakeholders included in the analysis it is quite likely that two or more scores will be the same, thus falling on top of each other. We solved this issue, in case of ties, by assigning one of the stakeholders ±0.5 on either importance or influence, depending on what was deemed most correct. Also, the limited number of importance-influence combinations resulted in stakeholders often being located on vertical or horizontal lines, thereby missing some of the delicate details of the analysis. In theory, this could be addressed by applying a psychometric scale that goes from, for example, 1 to 100, thus allowing for a higher resolution.

When applying the qualitative ranking approach, the stakeholders are assigned values from 1 to n, where n is the number of stakeholders included in the analysis. By doing this, some stakeholders are "forced" out to the borders of the importance-influence matrix. For the two other approaches, stakeholders, at least for the case of the microplastics restriction proposal, are placed less peripherally. Examples illustrating this point include how SMEs are forced toward the low-end boundary of the influence scale for the ranking approach, and how other national NGOs are forced to the low-end of the importance axis, Fig 1C. Also, the ranking approach does not allow stakeholders to be equally important or influential.

The arguments for how and why we scored the importance and influence of the different stakeholders as we did are the same regardless of the approach used. However, the analyses for each of the three approaches were done as independently of each other as possible, which in turn led to minor but still relevant differences in the results. For example, among others, UEFA scored lower on influence than regional authorities when assessing this vector on a scale from 1 to 10; however, for the two other approaches, it scored higher importance, Fig 1. These differences are most likely more attributable to the qualitative judgement of the authors than to the methods themselves. From the perspectives of the authors, doing top-down assessments of stakeholder importance and influence relying on expert judgement is associated with uncertainty. As mentioned, the approaches used herein were based on the same arguments, giving somewhat similar analyses, and even larger differences should be expected for an analysis done by different individuals or groups, each of whom offers different arguments for the scoring of stakeholder importance and influence. For this reason, there is a need to make the top-down importance-influence analysis step of the SA more robust to personal arguments and biases, more transparent in the derivation of the importance-influence scores and more objective, while at the same time maintaining the simplicity of the approach. Despite the differences described in the results of the three importance-influence matrix approaches, Fig 1, they are in consensus with regards to the overall trends. This comes as no surprise, as they are all built upon the same arguments.

The analysis of importance and influence indicates that just a handful of key stakeholders in the restriction proposal process is important for consideration. These include the ECHA, EC, EP, the Council, national governments and their environmental bodies, UN, CEFIC and large companies. In addition, it reveals that two stakeholders especially, namely SMEs and national sports NGOs, are vulnerable to the effects of the restriction proposal and should be

helped to adapt to the upcoming changes. This adaptation might take some time, though, which is why a proper transition period is important. By comparing these findings with the analyses of stakeholder activity, it is clear that the sport NGOs provide substantial input to the public hearing, whereas the SMEs are poorly represent. The sport NGOs will therefore be identified as a relevant stakeholder to include in the further policy process, but the SMEs are poorly represented and might thus be neglected, if the policy process solely relies on the public consultation.

## Top-down vs. bottom-up stakeholder analysis approach

When conducting a SA, it is important to remember that it is a snapshot of the present situation and not a static approach. Stakeholders' perceptions of things, their agendas, their interrelations and their importance and influence may change over time. In theory, a SA is an iterative process [22], which is essential for decision-makers to keep in mind when relying on the outcome. In practice, this means that SAs should always be used with a certain level of reservation. For instance, if important decisions build upon the perspectives of specific stakeholders, those stakeholders should be carefully monitored, to capture potential changes–and thus changes to the foundation of the decision. However, if the analysis is conducted on a solid platform, it might capture possible future changes, thus making it valid over a longer timeframe. In the case of the microplastics restriction proposal, the European Council, to date, has not been very involved. However, later in the process they will have to ratify it, thus becoming more important than depicted in our analysis.

When applying decision-support tools such as a SA with a top-down approach, the investigators conducting the analysis inevitably show bias. This is an inherent feature of qualitative decision-support tools relying on "expert" or "expert group" judgement, and it applies to our analysis as well [22]. One apparent strategy to overcome some of these inherent biases is to use a bottom-up approach, by reaching out to the stakeholders involved. In a perfect world, a bottom-up approach might be preferred to a top-down approach, but this is not always possible, and it is often both costly and time-consuming [21, 22].

As a result, there is a need for a more transparent and objective top-down approach to SAs. The literature especially is very vague on how to derive stakeholder importance and influence [19], but it is also challenging to assess importance and influence qualitatively, as the outcome is very dependent on the perspectives and arguments brought into play by the person(s) conducting the analysis. Thus, there is a need for an easy, transparent, top-down approach–quantitative or qualitative–for evaluating stakeholder importance and influence that is less sensitive to inherent biases. Ultimately, measuring stakeholder importance and influence is still the elusive goal. There is, however, great potential for the development of methods that combine qualitative top-down approaches with more detailed quantitative information in a manner that is more objective and less prone to investigator bias when it comes to the future development and implementation of stakeholder analysis. Our analysis points to some important policy implications that must be addressed in the future debate. First, the Annex XV revision process procedures have to be revised, if the EU governance strategy is to be fulfilled with respect to stakeholder inclusion. In the case of the Annex XV restriction proposal for intentionally added microplastics, the revised procedures must include policy measures to activate the public, the SMEs and representatives from academia in the debate. The limited contributions from academia stands in contrast with the ongoing scientific debate about risk of microplastic [85]. If restriction dossiers as Annex XV fail to account for scientific state of the art, it would have important policy implication. This becomes even more evident when analyzing the nature of the comments, where scientific arguments have a predominant role. The

inclusion of the SMEs might be challenging as we suspect that limited resources is the explanation for their poor engagement. Thus, the question of how to include those, who do not have the resources to engage, remains a challenge. Second, policy makers have to secure that vulnerable stakeholders can cope with the restrictions of the new Annex revisions. Again, the SMEs stands out and might require special policy solutions for the transition phase to ease the burden.

In conclusion, we find that the ECHA Annex XV restriction proposal process for inclusion of stakeholders did not capture all prominent stakeholders e.g. the public, SMEs and academics and researchers, and that there is a marked difference in the stakeholders that participate actively in the public consultation and that speak out in the media. The inclusion of stakeholders in the policy process can therefore not be based solely on the public consultation, as this would lead to the neglection of important stakeholders. Our findings suggest that an "advocacy coalition" has been formed between industry-related stakeholders and the national and international NGOs based on their opposition to the restriction proposal. Environmental and consumer NGOs, and national and international authorities are predominantly in favour of the restriction proposal, but do not seem to have aligned their arguments. Through our stakeholder analysis, we have identified environmental NGOs and SMEs as stakeholders with a lot at stake but limited power to influence. A common trait of all three methods for assessing stakeholder importance and influence is that they are top-down approaches relying on expert judgement. Thus, they tend to be subjective and reflects potential biases of the investigators. We conclude that a more objective top-down methodology to assess stakeholder importance and influence is needed. In light of the results of our stakeholder analysis, we recommend that:

- The ECHA should implement measures to include stakeholder groups, which, for various reasons, do not comment on the Annex XV restriction proposal. Our analysis shows that significant stakeholders are not represented in the public consultation. Primarily, academics, researchers and SMEs are poorly represented. Furthermore, the general public is not identified as an active stakeholder, neither in the Annex comments nor in the media. As stakeholder involvement is a cornerstone of EU regulation, these stakeholders are pivotal to include. This could be done, for instance, via the systematic use of new communication technologies that must be used to trigger and foster a new dialogue directly with the people in the EU as suggested by Van den Brande [6].

- RAC and SEAC should each invite 3–5 guest experts on the topic they are processing. Currently, it could seem that there is a lack of knowledge and documented expertise related to plastics among the committee members. A handful of guest experts would secure that state-of-the-art knowledge is available within the two ECHA committee and thus consolidate the scientific review and recommendations provided to ECHA.

- Academics and researchers should be more active during the public consultation process and not only in the media. Academia and researchers could have provided valuable insights and know-how on the scientific-technical issues and questions raised by the ECHA in the proposal, but we found that no representatives from this stakeholder group contributed to the process of providing comments on the restriction proposal.

- Special focus should be given to help SMEs cope with the restriction proposal. SMEs constitute the backbone of the European economy, but due to their position of low influence and high importance, they are vulnerable to the effects of the restriction proposal and need time to adapt.

- Media analysis is a good approach not only to identify important stakeholders but also as a mean to deduce the interests of the stakeholders. In our study, the niche media analysis

entailed identification of important stakeholders, which would otherwise have been excluded from the analysis.

- There is a need for a more objective measure of stakeholder importance and influence that combines qualitative top-down approaches with more detailed quantitative information, which at the same time maintain simplicity. Currently, available top-down approaches are too sensitive to bias and arguments brought into play by the people conducting the analysis and bottom-up approaches are too resource demanding for initial mapping of stakeholders.

## Supporting information

**S1 Checklist PRISMA 2009 checklist Clausen et al.**
(DOCX)

**S1 Table. Academia & researchers categorisation.**
(DOCX)

**S2 Table. Companies categorisation table.**
(DOCX)

**S3 Table. Industry and trade associations' categorisation table.**
(DOCX)

**S4 Table. International authorities categorisation table.**
(DOCX)

**S5 Table. International NGOs categorisation table.**
(DOCX)

**S6 Table. National authorities categorisation table.**
(DOCX)

**S7 Table. National NGOs categorisation table.**
(DOCX)

**S8 Table. SMEs categorisation table.**
(DOCX)

**S9 Table. International authorities microplastics comments.**
(DOCX)

**S10 Table. International NGOs microplastics comments.**
(DOCX)

**S11 Table. National authorities microplastics comments.**
(DOCX)

**S12 Table. National NGOs microplastics comments.**
(DOCX)

**S13 Table. Other contributors microplastics comments.**
(DOCX)

**S14 Table. Academia & researchers microplastics comments.**
(DOCX)

**S15 Table. Trade and industry associations microplastics comments.**
(DOCX)

**S16 Table. Companies Annex XV comments.**
(DOCX)

**S17 Table. Trade and industry associations Annex XV comments.**
(DOCX)

**S18 Table. International NGOs Annex XV comments.**
(DOCX)

**S19 Table. National NGOs Annex XV comments.**
(DOCX)

**S20 Table. Other contributors-authorities Annex XV comments.**
(DOCX)

## Acknowledgments

We thank MarinePlastic, a Danish Centre for research in marine plastic pollution, and the respective community for support and valuable input, making this research possible.

## Author Contributions

**Conceptualization:** Lauge Peter Westergaard Clausen, Nikoline Bang Oturai, Kristian Syberg, Steffen Foss Hansen.

**Data curation:** Lauge Peter Westergaard Clausen, Oliver Foss Hessner Hansen, Nikoline Bang Oturai, Kristian Syberg, Steffen Foss Hansen.

**Formal analysis:** Lauge Peter Westergaard Clausen, Oliver Foss Hessner Hansen, Nikoline Bang Oturai, Kristian Syberg, Steffen Foss Hansen.

**Methodology:** Lauge Peter Westergaard Clausen, Nikoline Bang Oturai, Kristian Syberg, Steffen Foss Hansen.

**Visualization:** Lauge Peter Westergaard Clausen, Oliver Foss Hessner Hansen, Steffen Foss Hansen.

**Writing – original draft:** Lauge Peter Westergaard Clausen, Kristian Syberg, Steffen Foss Hansen.

**Writing – review & editing:** Lauge Peter Westergaard Clausen, Oliver Foss Hessner Hansen, Nikoline Bang Oturai, Kristian Syberg, Steffen Foss Hansen.

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
