## [Decision Letter · Decision Letter 0]

3 Apr 2020

PONE-D-19-33681

Stakeholder analysis with regard to a recent European restriction proposal on microplastics

PLOS ONE

Dear authors,

Thank you for submitting your manuscript to PLOS ONE. After careful consideration, we feel that it has merit but does not fully meet PLOS ONE’s publication criteria as it currently stands. Therefore, we invite you to submit a revised version of the manuscript that addresses the points raised during the review process.

We would appreciate receiving your revised manuscript by Apr 20 2020 11:59PM. To enhance the reproducibility of your results, we recommend that if applicable you deposit your laboratory protocols in protocols.io, where a protocol can be assigned its own identifier (DOI) such that it can be cited independently in the future. For instructions see: http://journals.plos.org/plosone/s/submission-guidelines#loc-laboratory-protocols

We look forward to receiving your revised manuscript.

Kind regards,

Sabrina Gaito, Ph.D.

Academic Editor

PLOS ONE

Journal Requirements:

2.  As part of your revisions, please complete and submit a copy of the COREQ or SRQR checklists. Please ensure that you update your manuscript to include all the relevant information requested in the checklist.

3. Please ensure that the origin of the data is described in enough detail to be reproducible by other researchers, both in the Methods section and the Data availability statement.

"We thank MarinePlastic, a Danish Centre for research in marine plastic pollution sponsored by the VELUX foundation grant 25084, for providing the financial support for conducting this research."

"The authors received no specific funding for this work."

Reviewers' comments:

Reviewer's Responses to Questions

**Comments to the Author**

1. Is the manuscript technically sound, and do the data support the conclusions?

Reviewer #1: Yes

2. Has the statistical analysis been performed appropriately and rigorously? 

Reviewer #1: Yes

3. Have the authors made all data underlying the findings in their manuscript fully available?

Reviewer #1: Yes

4. Is the manuscript presented in an intelligible fashion and written in standard English?

Reviewer #1: Yes

5. Review Comments to the Author

Reviewer #1: This is an interesting paper. The topic is timely and follows appropriate guidelines related to the publication in a scientific journal. Nevertheless, the aim of the work is not clear. This is not clear in all the paper.

Moreover, the paper lacks of policy implication. Could be interesting added this part in the final section of the manuscript.

1. Line 21: avoid acronyms in the abstract.

2. Line 27: avoid acronyms in the abstract.

3. Line 88: add reference.

4. Line 93-94: modify with capital letters.

5. Line 101: specify acronym meaning.

6. Line 101-107: all throughout the introduction, it may seem that the focus of the paper is more on testing the top-down approach and the three classification methods, while here the MP issue appears to be the focus of the paper. This might be confusing. The authors should please provide a structure of the text in accordance with the priorities of the study.

7. Line 120: “stakeholders” here is a very general term. Moreover, not all the stakeholders identified later in the paper would need to directly comply with the restriction. Specify here which stakeholders will be involved in the peculiar circumstance.

8. Line 139: add reference.

9. Line 139-140: Is it the Authors’ opinion or ECHA opinion? It is not clear.

10. Line 153: lowercase.

11. Line 177: avoid repeating the title of the source, which is specified in the Reference section.

12. Line 178-179: avoid repeating the title of the source, which is specified in the Reference section.

13. Line 190: substitute “modified” with “adapted”.

14. Line 199: why the authors consider this specific timespan is not clear. Please specify.

15. Line 210: it is not clear what the authors mean by “groups” and what by “types”, and the difference between the two. The authors should please either specify or modify the sentence.

16. Line 214 to 218: It is not clear which is the scope of this classification within the paper. The authors should please further specify.

17. Line 257 to 269: the authors should please further specify the goal of the matrix within the study.

18. Line 312: the authors should please provide a brief comment on the advantages and disadvantages of the three methods, for completeness.

19. Line 327: the difference between a “stakeholder” and a “stakeholder subgroup” is unclear: the authors should please further specify.

20. Line 348: the authors should please specify the meaning of the acronym.

21. Line 373-374: the analysis of SMEs is here missing, although they have been identified as stakeholders commenting on the restriction proposal. The authors should please provide further analysis.

22. Line 457: spelling of “polymer” is wrong.

23. Line 461 to 467: the arguments seem to fall better in the “Criteria” category previously described by the authors. The authors should please either move the lines or specify why they fall under this category within the analysis.

24. Line 481 to 485: the arguments here described seem to relate more to costs for the companies than to scientific arguments. The authors should please either move the lines or specify why they fall under this category within the analysis.

25. Line 519: specify acronym meaning.

26. Line 535: the authors should please provide a brief description of the content of the “Others” category.

27. Line 588 to 589: the meaning of “importance” here seems discordant with the very precise definition of the term throughout the study that the authors gave in line 276. The authors should please rephrase the sentence or change term to avoid confusion.

28. Line 591: specify acronym meaning.

29. Line 602: specify acronym meaning.

30. Line 603-604: the authors should please move Table 3 after they have mentioned it in the text.

31. Line 605: specify acronym meaning.

32. Line 608: specify acronym meaning.

33. Line 611: specify acronym meaning.

34. Line 735 to 766: the authors should please move such methodological explanations to the appropriate section of the manuscript.

35. Line 777 to 779: the authors should please provide reference for such reasoning.

36. Line 828-829: the authors should please include a “Conclusion” section.

6. PLOS authors have the option to publish the peer review history of their article (what does this mean?). If published, this will include your full peer review and any attached files.

Reviewer #1: No

---

## [Author Response · Author response to Decision Letter 0]

20 Apr 2020

Dear Editor

We sincerely would like to thank the reviewer for a deliberate and thorough review, which have helped us refine and improve our manuscript. We have carefully taken all comments provided into consideration and revised the manuscript accordingly. Beneath, you find our responses to each of the comments provided and a description of the changes that we have made to the manuscript as a consequence of each comment. First, we address comments on the Journal Requirements. Second, we address the reviewer’s comments. Additionally, we have resubmitted the revised manuscript with and without track changes. 

Answers for comments on Journal Requirements:

Answer: Thanks. We have consulted the guidelines provided at the homepage of PLOS ONE and adjusted some mistakes throughout the manuscript. 

2. As part of your revisions, please complete and submit a copy of the COREQ or SRQR checklists. Please ensure that you update your manuscript to include all the relevant information requested in the checklist.

Answer: We completed a COREQ checklist to secure that all important aspects are included in the manuscript. We used the PRISMA template for systematic reviews, even though our study is not a traditional systematic review. The checklist is submitted as part of the revised submission. The checklist have not resulted in changes to the manuscript.

3. Please ensure that the origin of the data is described in enough detail to be reproducible by other researchers, both in the Methods section and the Data availability statement.

Answer: We have made some corrections to the Methods section, to leave no confusion about how we obtained all available data. 

Answer: We have no repository information. All data are available online and from our supporting information. We have stated this in the new cover letter as requested. 

"We thank MarinePlastic, a Danish Centre for research in marine plastic pollution sponsored by the VELUX foundation grant 25084, for providing the financial support for conducting this research."

"The authors received no specific funding for this work."

Answer: We have deleted the funding statement from the Acknowledgments. We would like to update the Funding Statement reading: We acknowledge MarinePlastic, a Danish Centre for research in marine plastic pollution, sponsored by the Velux Foundation, grant number 25084, for providing the funding for this research.

Answers to the reviewer’s comments:

Comments to the Author

1. Is the manuscript technically sound, and do the data support the conclusions?

Reviewer #1: Yes

Answer: Thank you.

2. Has the statistical analysis been performed appropriately and rigorously? 

Reviewer #1: Yes

Answer: Thank you. 

3. Have the authors made all data underlying the findings in their manuscript fully available?

Reviewer #1: Yes

Answer: Thank you. 

4. Is the manuscript presented in an intelligible fashion and written in standard English?

Reviewer #1: Yes

Answer: Thank you. 

5. Review Comments to the Author

Reviewer #1: This is an interesting paper. The topic is timely and follows appropriate guidelines related to the publication in a scientific journal. 

Answer: Thank you.

Nevertheless, the aim of the work is not clear. This is not clear in all the paper.

Answer: Thank you for highlighting this most important issue. We have revised the aims in the introductory section to stress, from the beginning, the purpose of our study. It now reads: “The aim of this study was to map the interests, influence and importance of active stakeholders in order to understand the arguments being put forward by different stakeholders and provide recommendations to policy-makers on how to ensure a balanced consideration of all stakeholder perspectives.”

Moreover, the paper lacks of policy implication. Could be interesting added this part in the final section of the manuscript.

Answer: Thanks. We have included policy implications in the last section (line 856-870). 

1. Line 21+ Line 27: avoid acronyms in the abstract.

Answer: We have removed all acronyms from the abstract 

3. Line 88: add reference.

 Answer: Thanks, we have inserted the reference. 

4. Line 93-94: modify with capital letters.

Answer: The abbreviation of the chemical industry branch organisation in the EU (CEFIC) is in French Conseil Européen des Fédérations de l'Industrie Chimique. We do not believe that it is meaningful to modify with capital letters in the English description/translation of CEFIC. 

5. Line 101: specify acronym meaning.

Answer: Thanks. We have spelled it out. 

6. Line 101-107: all throughout the introduction, it may seem that the focus of the paper is more on testing the top-down approach and the three classification methods, while here the MP issue appears to be the focus of the paper. This might be confusing. The authors should please provide a structure of the text in accordance with the priorities of the study.

Answer: We fully acknowledge that the aims of the paper were somewhat unclear. We have revised the aims-part to leave no confusion that the scope of the paper is to provide solid recommendations and suggestions for the future regulatory debate of microplastics. 

7. Line 120: “stakeholders” here is a very general term. Moreover, not all the stakeholders identified later in the paper would need to directly comply with the restriction. Specify here which stakeholders will be involved in the peculiar circumstance.

Answer: Thanks. We have provided an example of typical stakeholders, which need time to adapt to new regulations. 

8. Line 139: add reference.

Answer: Thank you. Reference has been added. 

9. Line 139-140: Is it the Authors’ opinion or ECHA opinion? It is not clear.

Answer: We have specified that it is ECHA’s opinion.

10. Line 153: lowercase.

Answer: Thanks. Corrected. 

11. Line 177-179: avoid repeating the title of the source, which is specified in the Reference section.

Answer: The title has been removed. 

13. Line 190: substitute “modified” with “adapted”.

Answer: “modified” has been substituted with “adapted”. 

14. Line 199: why the authors consider this specific timespan is not clear. Please specify.

Answer: We have specified that the first article published in the four online news media scrutinized was published back in 2013. Thus, the articles included are in our analysis are published in the mentioned period. 

15. Line 210: it is not clear what the authors mean by “groups” and what by “types”, and the difference between the two. The authors should please either specify or modify the sentence.

Answer: Thanks. We have modified the sentence by removing “types”. 

16. Line 214 to 218: It is not clear which is the scope of this classification within the paper. The authors should please further specify.

Answer: We have added a sentence explaining why it is important to do the categorization and what purposes it serves. It now reads: “The categorisation of stakeholders serves as a mean to secure that stakeholders on all levels are included in the analysis and provides an overview of the role and function of the stakeholders identified [46].”

17. Line 257 to 269: the authors should please further specify the goal of the matrix within the study.

Answer: Thanks, we have specified the purpose of the matrix. It now reads “The matrix is a tool to prioritise stakeholders [19] but also serve as a mean to identify potential vulnerable stakeholders – e.g. stakeholders with a lot at stake but limited power to influence.” 

18. Line 312: the authors should please provide a brief comment on the advantages and disadvantages of the three methods, for completeness.

Answer: We agree. We have added a short new section:

“A common trait of all three methods for assessing stakeholder importance and influence is that they, in this top-down approach, are relying on expert judgement. Thus, they are subjective approaches and reflects the biases of the investigators. It is important to note, that the “scale from low to high” methods is a qualitative assessment, whereas the two other methods are semi-qualitative, assigning a number to each of the stakeholders. Using a psychometric scale may lead to a somewhat ridged analysis, predefining the number of potential importance-influence combinations [53]. During the qualitative ranking approach, the stakeholders are assigned numbers based on their relative importance and influence. In principle, this entails that no stakeholders are equal in importance and influence and that e.g. the most influential stakeholder is located to the far right of the matrix, even though the stakeholder in reality only holds medium to high influence. These limitations and their influence on our analysis are further addressed in the discussion”.

19. Line 327: the difference between a “stakeholder” and a “stakeholder subgroup” is unclear: the authors should please further specify.

Answer: We agree. We have deleted the use of sub-groups from the manuscript.

20. Line 348: the authors should please specify the meaning of the acronym.

Answer: The acronym is specified on line 212 (218 in the revised manuscript). 

21. Line 373-374: the analysis of SMEs is here missing, although they have been identified as stakeholders commenting on the restriction proposal. The authors should please provide further analysis.

Answer: We agree and have revised the manuscript so that it now reads: 

“The limited number SMEs that do provide comments for the restriction proposal expresses anxiety due to the new requirements as well as for the new technical and administrative burden laid upon them.” 

22. Line 457: spelling of “polymer” is wrong.

Answer: We have corrected the spelling of polymer. 

23. Line 461 to 467: the arguments seem to fall better in the “Criteria” category previously described by the authors. The authors should please either move the lines or specify why they fall under this category within the analysis.

Answer: We agree and have moved the paragraph to the section on criteria. 

24. Line 481 to 485: the arguments here described seem to relate more to costs for the companies than to scientific arguments. The authors should please either move the lines or specify why they fall under this category within the analysis.

Answer: We agree and have added a new section titled “Other arguments” and moved the mentioned lines to this section. 

25. Line 519: specify acronym meaning.

Answer: Corrected 

26. Line 535: the authors should please provide a brief description of the content of the “Others” category.

Answer: We agree and have added a new section titled “Other arguments”. 

27. Line 588 to 589: the meaning of “importance” here seems discordant with the very precise definition of the term throughout the study that the authors gave in line 276. The authors should please rephrase the sentence or change term to avoid confusion.

Answer: We have rephrased the sentence and use “essential” instead. 

28. Line 591: specify acronym meaning.

Answer: We have specified the acronym. 

29. Line 602: specify acronym meaning.

Answer: We have specified the acronym. 

30. Line 603-604: the authors should please move Table 3 after they have mentioned it in the text.

Answer: We moved the table. 

31. Line 605: specify acronym meaning.

Answer: We have specified the acronyms. 

32. Line 608: specify acronym meaning.

Answer: We have specified the acronym. 

33. Line 611: specify acronym meaning.

Answer: We have specified the acronyms. 

34. Line 735 to 766: the authors should please move such methodological explanations to the appropriate section of the manuscript.

Answer: Unfortunately, we do not fully understand this comment. The lines referred to are (in our opinion) clearly belonging to the discussion. However, we believe that the reviewer might be referring to line 635-666. As a consequence, we have revised this paragraph so that it now reads: “As stakeholder involvement is considered one of the cornerstones of EU governance strategy [5], it is vital that all relevant stakeholder groups are either involved in the governance process or at least have been given a chance to participate. With respect to this, and to the ECHA’s Annex XV revisions, our analysis show several interesting aspects including, among others, how important stakeholders are not included in the Annex revision process and how stakeholders form advocacy coalitions”.

35. Line 777 to 779: the authors should please provide reference for such reasoning.

Answer: We have specified that this is our reasoning, building on our observations. 

36. Line 828-829: the authors should please include a “Conclusion” section.

Answer: We have added a concluding paragraph just before the recommendations so that it now reads: “In conclusion, we find that the ECHA Annex XV restriction proposal process for inclusion of stakeholders did not capture all prominent stakeholders e.g. the public, SMEs and academics and researchers, and that there is a marked difference in the stakeholders that participate actively in the public consultation and that speak out in the media. The inclusion of stakeholders in the policy process can therefore not be based solely on the public consultation, as this would lead to the neglection of important stakeholders. Our findings suggest that an “advocacy coalition” has been formed between industry-related stakeholders and the national and international NGOs based on their opposition to the restriction proposal. Environmental and consumer NGOs, and national and international authorities are predominantly in favour of the restriction proposal, but do not seem to have aligned their arguments. Through our stakeholder analysis, we have identified environmental NGOs and SMEs as stakeholders with a lot at stake but limited power to influence. A common trait of all three methods for assessing stakeholder importance and influence is that they are top-down approaches relying on expert judgement. Thus, they tend to be subjective and reflects potential biases of the investigators. We conclude that a more objective top-down methodology to assess stakeholder importance and influence is needed. In light of the results of our stakeholder analysis, we recommend that:”. 

6. PLOS authors have the option to publish the peer review history of their article (what does this mean?). If published, this will include your full peer review and any attached files.

Answer: It is fine with us to publish the peer review history.

---

## [Decision Letter · Decision Letter 1]

9 Jun 2020

Stakeholder analysis with regard to a recent European restriction proposal on microplastics

PONE-D-19-33681R1

Dear Dr. Clausen,

We’re pleased to inform you that your manuscript has been judged scientifically suitable for publication and will be formally accepted for publication once it meets all outstanding technical requirements.

Kind regards,

Sabrina Gaito, Ph.D.

Academic Editor

PLOS ONE

Additional Editor Comments (optional):

Reviewers' comments:

Reviewer's Responses to Questions

**Comments to the Author**

1. If the authors have adequately addressed your comments raised in a previous round of review and you feel that this manuscript is now acceptable for publication, you may indicate that here to bypass the “Comments to the Author” section, enter your conflict of interest statement in the “Confidential to Editor” section, and submit your "Accept" recommendation.

Reviewer #1: All comments have been addressed

2. Is the manuscript technically sound, and do the data support the conclusions?

Reviewer #1: Yes

3. Has the statistical analysis been performed appropriately and rigorously? 

Reviewer #1: Yes

4. Have the authors made all data underlying the findings in their manuscript fully available?

Reviewer #1: Yes

5. Is the manuscript presented in an intelligible fashion and written in standard English?

Reviewer #1: Yes

6. Review Comments to the Author

Reviewer #1: (No Response)

7. PLOS authors have the option to publish the peer review history of their article (what does this mean?). If published, this will include your full peer review and any attached files.

Reviewer #1: No

---

## [Editor Report · Acceptance letter]

11 Jun 2020

PONE-D-19-33681R1 

Stakeholder analysis with regard to a recent European restriction proposal on microplastics 

Dear Dr. Clausen:

I'm pleased to inform you that your manuscript has been deemed suitable for publication in PLOS ONE. Congratulations! Your manuscript is now with our production department. 

Kind regards, 

on behalf of

Dr. Sabrina Gaito 

Academic Editor

PLOS ONE